# Identification and Characterization of NPR1 and PR1 Homologs in *Cymbidium* orchids in Response to Multiple Hormones, Salinity and Viral Stresses

**DOI:** 10.3390/ijms21061977

**Published:** 2020-03-13

**Authors:** Rui Ren, Yonglu Wei, Sagheer Ahmad, Jianpeng Jin, Jie Gao, Chuqiao Lu, Genfa Zhu, Fengxi Yang

**Affiliations:** Guangdong Key Laboratory of Ornamental Plant Germplasm Innovation and Utilization, Environmental Horticulture Research Institute, Guangdong Academy of Agricultural Sciences, Guangzhou 510640, China; renruinjau@163.com (R.R.); sagheerhortii@gmail.com (S.A.);

**Keywords:** *Cymbidium* orchid, Genome-wide identification, Hydrogen peroxide (H_2_O_2_), *Nonexpressor of pathogenesis-related 1* (*CsNPR1*), *Pathogen-associated 1* (*CsPR1*), Salicylic acid (SA)-dependent, *Cymbidium* mosaic virus (CymMV)

## Abstract

The plant *nonexpressor of pathogenesis-related 1* (*NPR1*) and *pathogenesis-associated 1* (*PR1*) genes play fundamental roles in plant immunity response, as well as abiotic-stress tolerance. Nevertheless, comprehensive identification and characterization of *NPR1* and *PR1* homologs has not been conducted to date in *Cymbidium* orchids, a valuable industrial crop cultivated as ornamental and medicinal plants worldwide. Herein, three *NPR1*-like (referred to as *CsNPR1-1*, *CsNPR1-2*, and *CsNPR1-3*) and two *PR1-like* (*CsPR1-1* and *CsPR1-2*) genes were genome-widely identified from *Cymbidium* orchids. Sequence and phylogenetic analysis revealed that *CsNPR1-1* and *CsNPR1-2* were grouped closest to *NPR1* homologs in *Zea mays* (sharing 81.98% identity) and *Phalaenopsis* (64.14%), while *CsNPR1-3* was classified into a distinct group with *Oryza sativa*
*NPR 3* (57.72%). CsPR1-1 and CsPR1-2 were both grouped closest to *Phalaenopsis* PR1 and other monocot plants. Expression profiling showed that *CsNPR1* and *CsPR1* were highly expressed in stem/pseudobulb and/or flower. Salicylic acid (SA) and hydrogen peroxide (H_2_O_2_) significantly up-regulated expressions *of CsNPR1-2*, *CsPR1-1* and *CsPR1-2,* while *CsNPR1-3*, *CsPR1-1* and *CsPR1-2* were significantly up-regulated by abscisic acid (ABA) or salinity (NaCl) stress. *In vitro* transcripts of entire *Cymbidium* mosaic virus (CymMV) genomic RNA were successfully transfected into *Cymbidium* protoplasts, and the CymMV infection up-regulated the expression of *CsNPR1-2*, *CsPR1-1* and *CsPR1-2.* Additionally, these genes were transiently expressed in *Cymbidium* protoplasts for subcellular localization analysis, and the presence of SA led to the nuclear translocation of the CsNPR1-2 protein, and the transient expression of *CsNPR1-2* greatly enhanced the expression of *CsPR1-1 and CsPR1-2*. Collectively, the *CsNPR1-2*-mediated signaling pathway is SA-dependent, and confers to the defense against CymMV infection in *Cymbidium* orchids.

## 1. Introduction 

The *Orchidaceae* is one of the largest and the most evolved families of monocot plants [1]. More than 70,000 orchids have been cultivated as ornamental and medicinal plants throughout the world [2]. *Cymbidium* orchids, best known for their aesthetic appeal and ideal characteristics, are widely distributed and popular in subtropical and tropical Asia [3,4]. However, numerous biotic (fungal, bacterial and viral diseases) and abiotic stresses (drought, salinity, etc.) seriously affect orchid production [5]. For instance, *Cymbidium* mosaic virus (CymMV, *Potato virus X*) and *Odontoglossum* ringspot virus (*Tobamovirus*) are two prevalent viral pathogens infecting orchids that cause chlorosis, necrosis, and dwarfing symptoms on orchid plants, thereby reducing the ornamental and economic value of orchids [6]. Replacing infected plants with virus-free germchit generated through meristem culturing has been widely applied as the main control measure for orchid viral diseases [7,8]. However, orchid tissue culturing is costly, labor-intensive and time-consuming. The long vegetative growth period of orchids also makes it necessary to find a permanent solution for this issue. The introduction of natural and broad-spectrum resistance against pathogens is regarded as the most economical and eco-friendly strategy for plant disease management [9,10,11]. 

The nonexpressor of pathogenesis-related 1 (NPR1) plays a key role in multiple signaling pathways in plant immunity [12] and abiotic-stress tolerance [13,14,15]. More concretely, the NPR1 protein has been demonstrated as an essential regulator of long-lasting and broad-spectrum systemic acquired resistance (SAR) contributing to resistance against fungal, bacterial and viral pathogens [16,17]. SAR is induced by systemic changes in endogenous molecules, such as a burst of reactive oxygen species (ROS) and changes in hormonal levels following local infection of pathogens [18,19]. When SAR occurs, several *pathogenesis-related* (*PR*) genes are active locally at the site of infection and systemically in distal plant tissues [20,21]. The inducible SAR confers broad-spectrum immunity not only to secondary infection but also to other pathogens [19,22,23]. Since the first NPR1 protein was identified in *Arabidopsis* through a mutant screen [24], *NPR1-like* homologs have been identified from various plant species (Appendix A). However, the identification of plant immunity is mostly limited to model plants, including *Arabidopsis thaliana,* rice (*Oryza sativa*) and tobacco (*Nicotiana spp.*). For the species’ limitations, little progress on orchid defense and/or tolerance to these pathogenesis stresses has been reported. Chen et al. first identified the orchid *PhaNPR1-like* and *PhaPR1-like* genes from *Phalaenopsis* [25], and a C3HC4-type RING-finger domain-containing transcription factor *PhaTF15* was shown to be involved in the regulation of *PhaNPR1*, *PhaPR1*, and virus accumulation [26]. Nevertheless, *NPR1-like* or *PR1-like* genes have not yet been identified from *Cymbidium* orchids. 

Constitutive expression of *NPR1* within wild-type *Arabidopsis thaliana* ensures a quick response to salicylic acid (SA) [27]. Recent research has demonstrated that NPR1 transduces the SA signal to activate expression of PR genes [28]. NPR1 protein is predominantly oligomeric and sequestered in the cytoplasm in the absence of pathogen infection [16]. Pathogen infections induce the accumulation of SA in both inoculated and systematic leaves of plants. Meanwhile, biphasic change in cellular reduction potentially results in reduction in NPR1 to a monomeric form following pathogen challenges. Acting as a receptor of SA, monomeric NPR1 protein directly binds SA and translocates to the nucleus [29], which is required for the activation of PR gene expression [30]. However, whether the NPR1-dependent SA signaling in orchid plants is conserved in other species remains unclear. 

For the first time, we carried out a genome-wide identification of *NPR1* and *PR1* homologs in *Cymbidium* orchids. To investigate the involvement of the NPR1-mediated signaling pathway in *Cymbidium* orchids, the expression of *CsNPR1* and *CsPR1* genes in response to phytohormones, abiotic salinity stresses and CymMV infection were examined. Additionally, *CsNPR1 and CsPR1* genes were transiently expressed in the *Cymbidium* protoplasts for subcellular localization and gene regulation analysis. Finally, we figured out the NPR1-mediated signaling pathway, which is Salicylic acid-dependent and triggered by CymMV infection in *Cymbidium* orchids. 

## 2. Results

### 2.1. Genome-Wide Identification of NPR1 and PR1 Homologs in Cymbidium Orchids

To identify *NPR1* and *PR1* homologs in *Cymbidium* orchids, a local protein BLAST search of homolog sequences from *Phalaenopsis* and rice was performed against unpublished *C. sinense* database. Genome-wide identification resulted in three *CsNPR1*-like (designated *CsNPR1-1*, *CsNPR1-2* and *CsNPR1-3*) and two *CsPR1*-like genes (*CsPR1-1* and *CsPR1-2*) being identified from *Cymbidium sinense*. In the full genomic sequence of each *CsNPR1* gene, there were three introns and four exons, but only one exon exists in each *CsPR1* gene (Appendix A). Sequence analysis revealed that the coding sequences (CDSs) of *CsNPR1-1*, *CsNPR1-2*, *CsNPR1-3*, *CsPR1-1* and *CsPR1-2* contained 1707, 1668, 1446, 501 and 504 -bp open reading frames, encoding putative proteins of 569, 556, 482, 167 and 168 amino acids, respectively (Figure 1). Domain architecture analysis revealed that CsNPR1 proteins shared conserved broad-complex, tramtrack, bric-a-brac/poxvirus and zinc finger (BTB/POZ) domains, two or three ankyrin repeat (ANK) domains, and an NPR1-like defense protein C domain with other NPR1 homologs (Figure 1A). Putative CsPR1 proteins share a conserved signal peptide (SP) and a sterol carrier protein (SCP) domain with PR1-like proteins in other plant species (Figure 1B). 

### 2.2. Phylogenetic Analysis and Sequence Alignment of NPR1 and PR1 Homologs

To investigate the homology between NPR1-like and PR1-like proteins, phylogenetic analysis and sequence alignments were carried out using their full-length deduced amino acid sequences (Appendix A). Among the three CsNPR1-like homolog proteins, CsNPR1-1 and CsNPR1-2 showed a higher homology (sharing 67.57% identity), and were the group closest with NPR1 proteins in *Zea mays* (sharing 81.98% identity) and *Phalaenopsis* (64.14% identity), respectively (Figure 2A). However, CsNPR1-3 was classified into a distinct group with the *Calotropis procera* NPR1 (70.68% identity) and *Oryza sativa* NPR1-like protein 3 (NPR3) (57.72% identity), respectively. The BTB/POZ domains of NPR1-like proteins are highly conserved (Appendix A). In addition, PR1-like proteins were classified into two major groups, and CsPR1-1 and CsPR1-2 proteins were grouped together with other PR1 proteins from monocots (Figure 2B). Putative CsPR1-1 and CsPR1-2 proteins were highly conserved (99.40% identity), and formed a distinct subgroup with the *Phalaenopsis* PR1 protein, sharing 68.67% and 68.26% identity, respectively (Figure 2B and Appendix A). 

### 2.3. Tissues/Organs-Specific Expression Analysis of CsNPR1 and CsPR1 Genes

To characterize the expression patterns of *CsNPR1* and *CsPR1* genes, the expression profiles of these genes in different tissues/organs (roots, stem/pseudobulb, leaf, flower and pod) (Figure 3A) were evaluated by quantitative reverse-transcription polymerase chain reaction (qRT-PCR). Although they had different relative expression levels, these genes were detectable in various tissues/organs. *CsNPR1-1* and *CsNPR1-2* showed similar expression patterns regardless of their overall gene expression levels, with the highest expression levels in the stem/pseudobulb followed by flower, while *CsNPR1-3* showed the highest expression in flower (Figure 3B). The relative expression levels of *CsNPR1-1* and *CsNPR1-3* were much lower than that of *CsNPR1-2* overall, which indicate that *CsNPR1-2* is more active. The expression patterns of the two *CsPR1* genes were similar, and both *CsPR1-l* and *CsPR1-2* showed the highest expression levels in flower (Figure 3C). The results showed that *CsNPR1* and *CsPR1* genes were constitutively expressed in various tissues/organs, with higher expression levels in the plant aerial portion than in others. 

### 2.4. SA and ROS Significantly Upregulates Expression of CsNPR1 and CsPR1 Genes

To characterize the expression patterns of *CsNPR1* and *CsPR1* genes, their response to various inductions were investigated in *Cymbidium* protoplasts. As shown in Figure 4, protoplasts were efficiently isolated from *Cymbidium* flower petals. The maximum yield and viability reached to approximately 3.3 × 10^7^/g (fresh weight) and 92.3%, respectively. Following protoplast treatments with various phytohormones and stresses, the relative expression levels of *CsNPR1* and *CsPR1* genes were measured by qRT-PCR. 

Protoplast treatments were simultaneously carried out, which is more reliable, repeatable and precise than using plants. With the addition of 100 -μM SA, *CsNPR1* and *CsPR1* genes exhibited similar inducible expression patterns, and their expression levels were greatly increased at the early stage (6 hpt), and decreased gradually as time passed, when compared with controls at each timepoint (6, 12, and 24 hpt) (Figure 5A). Moreover, *CsNPR1-2* showed greater fold changes (almost 20-40 folds) compared with other genes, which showed almost equal fold changes (3 to 4 folds). Meanwhile, the protoplast treated with 200 -μM Abscisic acid (ABA) was conducted (Figure 5A). Different from their responses to SA, only *CsNPR1-3* was significantly up-regulated by ABA, while *CsNPR1-1* and *CsNPR1-2* showed no significant difference compared to the control at all three timepoints. Interestingly, *CsPR1-1* and *CsPR1-2* were also up-regulated by ABA at 6 and/or 12 hpt. *Cymbidium* protoplasts were also treated with hydrogen dioxide peroxide (H_2_O_2_) and ascorbic acid (AsA), respectively. ASA is a water-soluble anti-oxidant molecule which acts as a primary substrate in the cyclic pathway of enzymatic detoxification of H_2_O_2_ [31]. The expressions of *CsNPR1* and *CsPR1* genes were significantly up-regulated in the presence of H_2_O_2_, but were remarkably inhibited with the added AsA when compared with controls at each timepoint (Figure 5B). Additionally, the protoplasts were also treated with 200 -mM NaCl salinity stress. It was found that the expression levels of *CsNPR1-1* and *CsNPR1-3* were significantly up-regulated, while those of *CsNPR1-2* showed few differences between the experimental and control groups (Figure 5C). The expressions of *CsPR1* genes were only significantly up-regulated at 6 hpt by salinity stress. Taken together, SA and ROS, but not ABA or salinity stresses, greatly triggered the expression of *CsNPR1* and *CsPR1* genes, and *CsNPR1-2* was most responsive.

### 2.5. CsNPR1-2-Mediated Singling Pathway Confers the Response of Cymbidium orchids Against CymMV Infection

To explore the expression patterns of the defense-related genes *CsNPR1* and *CsPR1* in response to virus infection, the *in vitro* transcripts of entire CymMV-RNA and blank control (equal volumes of water) were transfected into *Cymbidium* protoplasts. Fluorescein diacetate (FDA) staining was used to determine protoplast viability at each timepoint. It was shown that most protoplasts remained viable after transfection (Figure 6A). The replication of CymMV in transfected protoplasts was detected by double-antibody sandwich-enzyme linked immunosorbent assay (DAS-ELISA) and RT-PCR. After 24 h of cultivation, CymMV replication was detectable in the protoplasts. The target-fragments of the CymMV-coat protein (CymMV-CP) coding sequence were amplified from transfected protoplasts at 24 hpt, while no product was amplified from the negative control sample (Figure 6B). Additionally, yellow coloration was observed in the DAS-ELISA wells representing CymMV infection of transfected protoplasts, and the OD_405 nm_ values exceeded three times that of the negative-control wells (Figure 6C). All these results indicated CymMV infection in the transfected *Cymbidium* protoplasts. 

Subsequently, the expression patterns of CsNPR1 and CsPR1 genes in response to CymMV infection in Cymbidium protoplasts were examined at 6, 12 and 24 hpt. As shown in Figure 6D, the expression of CsNPR1-2 was significantly up-regulated at 12 and 24 hpt compared with blank control, while that of CsNPR1-1 was slightly up-regulated and CsNPR1-3 was down-regulated at 24 hpt, respectively. Meanwhile, both CsPR1-1 and CsPR1-2 were greatly up-regulated by 4-6-fold from 12 to 24 hpt. Therefore, CsNPR1-2, rather than CsNPR1-1 and CsNPR1-3, conferred the orchid-immunity, and the NPR1-2-mediated singling pathway was involved in the Cymbidium response against CymMV infection. 

### 2.6. Nuclear Translocation of CsNPR1-2 in the Presence of SA

To characterize the *CsNPR1* and *CsPR1* homologous proteins, they were transiently expressed in *Cymbidium* protoplasts for protein subcellular localization. Recombinant vectors expressing fusion proteins CsNPR1-1::GFP, CsNPR1-2::GFP CsNPR1-3::GFP, CsPR1-1::GFP and CsPR1-2::GFP, and an empty vector only expressing green fluorescent protein (GFP), were transformed into *Cymbidium* protoplasts (Figure 7A). Strong green fluorescence was observed in most protoplasts at 16-24 hpt, and a maximum transfection efficiency greater than 80% was achieved according to the ratio of protoplasts with fluorescence. Fusion proteins CsNPR1-2::GFP and CsNPR1-3::GFP were distributed throughout the entire cell, which was similar to that of the vector control, while CsNPR1-1::GFP, CsPR1-1::GFP and CsPR1-2::GFP were mainly expressed at the nucleus and plasma membrane (Figure 7B). In addition, the fusion protein CsNPR1-2::GFP accumulated in the nucleus in the present of 100 μM SA (Figure 7B). However, adding SA did not change the subcellular localizations of fusion proteins CsPR1-1::GFP and CsPR1-2::GFP. These results indicated that the presence of SA induced the nuclear translocation of CsNPR1-2 protein. 

### 2.7. CsNPR1-2 Positively Regulate the Expression of CsPR1 Genes

To profile the *NPR1*-mediated singling pathway in *Cymbidium* orchids, the regulation of the strong candidate CsNPR1-2 protein on expression of *CsPR1* genes was investigated. Protoplasts transiently expressing fusion protein CsNPR1-2::GFP were collected at 6, 12 and 24 hpt, and expressions of *CsPR1-1* and *CsPR1-2* genes were examined using qRT-PCR. Fusion protein CsNPR1-2::GFP was successfully expressed with bright green fluorescence in most *Cymbidium* protoplasts at 24 hpt. The expression levels of *CsNPR1-2* reached 200-2500 fold during 6-12 hpt (Figure 8). Meanwhile, the expressions of *CsPR1* genes were greatly up-regulated throughout 6-24 hpt, but excess *CsNPR1-2* accumulation led to the decreased expression of *CsPR1* genes. Despite *CsPR1-1* and *CsPR1-2* showing similar expression patterns, the relative expression level of *CsPR1-2* was almost 10 -times that of *CsPR1-1*. These results indicated that transient expression of *CsNPR1-2* protein greatly enhanced the expression of *CsPR1* genes in *Cymbidium* orchids. 

## 3. Discussion

Plant pathogens seriously affect the global production of crops and horticultural species [32,33]. Reasonable disease management approaches are eco-friendly and desirable to limit production inputs and provide economic benefits to farmers [33]. Plant defense/immunity mechanisms against various pathogens have been studied extensively in model plants and main crops. However, insufficient genomic information and laggard biotechnology greatly limit the discovery of *R*-genes in horticultural species. The identification and application of key homologs has been proven an efficient method to generate disease-resistant transgenic plants [34,35,36,37,38,39]. Heterologous expression of *AtNPR1* in horticultural crops such as tomato [35], carrot [40] and citrus [37] shows the potential to generate transgenic plants with increased broad-spectrum disease and pest resistance. Herein, we identified three *NPR1*-like and two *PR1*-like homologs from *Chinese Cymbidium* orchids, which are splendid ornamental plants worldwide. A detailed characterization and utilization of these plant defense-related homologs would facilitate disease-resistance breeding in *Orchidaceae* and other plant species. 

Previously, full-length transcripts of *PhaNPR1* and *PhaPR1* were obtained with the primers designed from the conserved domain sequences and the rapid amplification of cDNA ends [25]. However, the reported *NPR1*-like families in other plant species are comprised of several members, for instance, six *NPR1-like* genes (*AtNPR2*, *AtNPR3*, *AtNPR4*, *AtNPR5*/*AtBOP1* and *AtNPR6*/*AtBOP2*) have been described in *Arabidopsis* [41]. Hence, we inferred that there should be more members of the *NPR1-like* family in the orchid genome with potentially sub-functional differentiation. In this study, a genome-wide identification of *NPR1* and *PR1* homologs was carried out against an unpublished *Cymbidium* orchid genome dataset. A total of three *CsNPR1*-like and two *CsPR1*-like genes were identified from *C. sinense* (Figure 1). For these putative *Cymbidium* NPR1-like proteins, CsNPR1-1 showed the closest homology with ZmNPR1 from *Zea mays* (sharing 81.98% identity), but CsPR1-2 with PhaNPR1 showed the closest homology with the *Phalaenopsis* orchid (64.14% identity) (Figure 2). As one of the most evolved families of monocot plants [1], the genomic organization of *CsNPR1-like* and *CsPR1-like* families in orchids might be more complex than in other plant species. The NPR1 plays a key role in multiple signaling pathways in plant immunity [12], as well as in plant development [42] and abiotic-stress tolerance [13,14,15]. Moreover, *AtNPR1* and *AtNPR4* exhibit opposite roles in early defense gene expression in response to SA [43]. In the present study, *CsNPR1-2* showed different expression patterns to *CsNPR1-1*/*3* (Figure 4, Figure 5 and Figure 6), indicating that they probably have different roles in the NPR1-dependent singling pathway. 

The responses of plants to pathogen infection, and other biological and abiotic stresses, involve changes in ROS and hormone levels [44,45,46,47]. This led us to investigate the involvements of *CsNPR1* and *CsPR1* homologs in response to various inductions, including by SA, H_2_O_2_, ABA, salinity (NaCl) stress and CymMV infection. Although *CsNPR1* and *CsPR1* genes maintained a low expression level in healthy plants (Figure 3B), they were significantly up-regulated by the inductions. Briefly, the presence of SA significantly up-regulated the expression of *CsNPR1* genes at 6 hpt (Figure 5A), and caused the nuclear translocation of the CsNPR1-2 protein (Figure 7B). These results indicated that CsNPR1-2 was involved in the response and transduction of SA signals in *Cymbidium* orchids (Figure 8). Though plant NPR1 regulates the expression of *PR1-like* genes via the interaction with basic leucine zipper protein transcription factors [48,49,50,51], exogenous SA and the transient expression of *CsNPR1-2* enhanced the expression of *CsPR1-1* and *CsPR1-2* (Figure 5A and Figure 8). Moreover, the expression levels of *CsPR1* genes upregulated by *CsNPR1-2* overexpression were much higher than those upregulated by SA, indicating that SA triggered the expression of *CsPR1* genes through the promotion of *CsNPR1-2* expression. In conjunction with SA and nitric oxide, ROS primarily function as signal transduction molecules in crosstalk among different pathways [52,53]. Herein, the expressions of *CsNPR1-2, CsPR1-1* and *CsPR1*-2 were greatly up-regulated by H_2_O_2_ induction and CymMV infection (Figure 5 and Figure 6). Plant viral infection usually results in an ROS burst and hypersensitive necrosis [54,55], but this requires further verification with regard to the CymMV infection of orchid plants. It was inferred that the defense response of *Cymbidium* orchids against CymMV infection probably started with the ROS burst and accumulated SA, which in turn activated the *CsNPR1-2* expression and *CsPR1* genes (Figure 9).

Traditionally, plant viruses can be easily mechanically inoculated into dicot hosts, such as soybean, tobacco and cucumber [56]. However, it is very difficult to mechanically inoculate the virus into the leathery leaves of orchids and some other monocot plants for plant–virus interaction study. Hence, *in-vitro*-transcribed entire viral genomic RNA or viral infectious clone vectors were transformed into protoplasts, which enables virus replication [57,58,59] and virus–plant interaction studies [60,61,62]. In this study, *Cymbidium* protoplasts were transfected with *in-vitro*-transcripted entire CymMV-RNA, and CymMV were detectable in the transfected protoplast at 24 hpt (Figure 6B,C). The viral replication in protoplasts means that the virus utilizes the transcription and translation systems of plant cells [63]. The plant–virus interaction triggered the expression of *CsNPR1* and *CsPR1* genes (Figure 6D), indicating that our protoplast-based viral-RNA transfection method can be successfully used for orchid–virus research.

## 4. Materials and Methods

### 4.1. Plant Materials

Healthy virus-free *Cymbidium sinense* plants were used. These orchid plants were obtained from the orchid breeding base of Environmental Horticulture Research Institute, Guangdong Academy of Agricultural Sciences, China, and maintained in plastic pots (20 × 20 cm) in greenhouses with a favorable environment for growing as previously described [64]. 

### 4.2. Genome-Wide Identification of NPR1-like and PR1-like Genes in Cymbidium Sinense

The sequences of *PhaNPR1* in *Phalaenopsis aphrodite* (GenBank accession number: JN630802.1) and *OsNPR1* in rice (DQ450947.1) (Appendix A), as well as *PhaPR1* in *P. aphrodite* (JX137044.1) and *OsPR1* in rice (AF306651.1) were retrieved from the National Center for Biotechnology Information (NCBI) GenBank (https://www.ncbi.nlm.nih.gov) database (Appendix A). A local protein database of *Cymbidium sinense* (unpublished) was established for the basic local alignment search (BLAST, ftp://ftp.ncbi.nlm.nih.gov/blast/executables/blast+/LATEST/). These deduced protein homolog sequences were used as queries to search for NPR1-like and PR1-like proteins in the protein database of *Cymbidium sinense* (E-value < 1 × 10^−5^). Candidate sequences were retrieved and verified for conserved domains using the Pfam database (https://pfam.xfam.org/) and InterPro database (http://www.ebi.ac.uk/interpro/), and candidate proteins without conserved domains were removed. Finally, the genes encoding the putative NPR1-like and PR1-like proteins were identified from the unpublished *Cymbidium sinense* genome. Exon-intron structures of these identified genes were predicted with the GENSCAN software (http://genes.mit.edu/GENSCAN.html). The CDSs of *CsPR1-like* and *CsNPR1-like* homologs were identified according to our previous *De novo* transcriptome assembly data [65].

### 4.3. Sequence and Phylogenetic Analysis

Total RNA was extracted from the young leaves of *Cymbidium sinense* with a RNA Simple Total RNA Kit (Tiangen, Beijing, China). The DNA-free RNA was used for first-strand cDNA synthesis using Oligo (dT) primers and a PrimeScript™ И 1st strand cDNA Synthesis Kit (Takara, Dalian, China) following the manufacturer’s instructions. Gene-specific primers were designed according to the CDSs of *CsPR1-like* and *CsNPR1-like* suing Primer Premier 5.0 software (Premier, Palo Alto, CA, USA) (Appendix A). Subsequently, the fragments for each gene were amplified from the cDNA using PrimerSTAR Max DNA Polymerase (Takara, Dalian, China). Purified products were cloned into the pMD18-T vector (Takara, Dalian, China) and sequenced. The obtained CDSs of *CsNPR1-like* and *CsPR1-like* genes were analyzed for ORFs using GENESCAN software. The domains and architectures of putative proteins were predicted using online software SMART (http://smart.embl-heidelberg.de) and InterProScan (http://www.ebi.ac.uk/InterProScan). 

Phylogenetic analysis and sequence alignments were conducted with the full-length deduced amino acid sequences of NPR1-like and PR1-like proteins (Appendix A). The phylogenetic trees were generated using the MEGA 5.10 software with the neighbour-joining method applying 1000 replicates [66]. According to the results of phylogenetic analysis, homologs in other species having close phylogenetic relationships were aligned and analyzed for protein identities using Geneious Prime v. 2019.0.3 (BioMatters, Ltd., Auckland, New Zealand). 

### 4.4. Tissues Collection and Protoplast Isolation

Various tissues including root tip, stem (protocorm), leaf, flower petal and immature pod were collected from healthy *Cymbidium sinense* plants. Total RNA was extracted, and DNA-free RNA was reverse-transcribed as mentioned above. The expression levels of *CsNPR1-like* and *CsPR1-like* genes in various tissues/organs were examined by quantitative real-time polymerase chain reaction (qRT-PCR).

Moreover, the fully expanded flower petals were also collected for protoplast isolation. The protoplast isolation was conducted following the past protocols [67,68]. Generally, petals were cut into 0.5~1.0 mm strips and transferred into the freshly prepared enzyme-solution (1.0% (weight/volume, *w*/*v*) Cellulase R10, 0.5% (*w*/*v*), Macerozyme R10, 500-mM D-mannitol, 20-mM KCl, and 20-mM MES (pH = 5.7), 10-mM CaCl_2_, 0.1% (*w*/*v*) BSA). The released protoplasts were harvested after incubation at 28 °C in the darkness with rotations of 30 rpm for 5–6 h. The purified protoplasts were adjusted to a suitable density with W5 solution (154 mM NaCl, 125 mM CaCl_2_, 5 mM KCl and 2 mM MES, pH 5.7). 

### 4.5. Protoplast Measurements and Treatments

Protoplasts were counted and photographed with a hemocytometer under a Leica DM2500 microscope (Leica, Wetzlar, Germany). FDA staining was used to determine protoplast viability [69], under a LSM 710 confocal laser microscope (Carl Zeiss, Inc., Jena, Germany). The different treatments of protoplasts were carried out in W5 solution supplemented with 100 μM SA, 500 μM H_2_O_2_, 2 mM AsA, 200 μM ABA and 100 mM NaCl. The optimum concentrations of treatments were determined according to those for other plants, such as *Arabidopsis* [53], rice [70], *Phalaenopsis* [70] and soybean [71]. Approximately 10^6^ protoplasts in 1 mL W5 solution were used for each treatment. W5 solution was supplemented with water as the control treatment at each timepoint. Protoplasts were cultivated in 6-well plates in a growth chamber at 23 °C for 6–24 h in the darkness. 

### 4.6. In Vitro Transcription of CymMV-RNA and Vector Preparation 

The viral RNA was generated from plasmid p18Cy13, a biologically active cDNA clone of CymMV encompassing the entire viral RNA of CymMV and T7 RNA polymerase promoter fused to the 5′ extreme of the viral cDNA [72]. Plasmid p18Cy13T7 (gift from Prof. Sek-Man Wong, National University of Singapore) was linearized with *Sma*I, and the digested DNA was purified and transcribed using the mMESSAGE mMACHINE^TM^ T7 Ultra kit (Ambion mMessage mMachine, Austin, TX, USA) following the manufacturer’s instructions. *In vitro* transcripts (15 μg) were then transfected into *Cymbidium* protoplasts. 

The vectors used for protoplast-based transient expression were obtained by inserting the CDSs of these genes into the vector PAN580-GFP. The gene-specific primers with overlapping homologous ends were designed using Primer Premier 5.0 software (Premier, Palo Alto, CA, USA) according to the full-length CDSs of *CsNPR1* and *CsPR1* genes (Appendix A). Subsequently, the fragments for each gene were amplified and purified, and full-length CDS without a termination codon were cloned into the PAN580-GFP vector by recombination using the Seamless Assembly Cloning Kit (CloneSmarter, Houston, TX, USA) following the manufacturer’s instructions. Recombined vectors were transformed into *Escherichia coli* DH5α-competent cells (TianGen, Beijing, China) according to the manufacturer’s instructions and confirmed by sequencing. Following mass replication of the bacterium, plasmid DNA was extracted by Endo-Free Plasmid Maxi Kit (Omega Bio-tek, Norcross, GA, USA). The concentrated plasmid DNA (2.0 μg/μL) was prepared for protoplast transfection. 

### 4.7. PEG-Mediated Protoplast Transfection

Protoplast transfection was carried out using the PEG-mediated protocol with minor modifications [67]. Briefly, an equal volume of freshly prepared PEG solution (40% (*w*/*v*) PEG 4000, 0.2 M mannitol and 0.1 M CaCl_2_) was gently mixed with pre-assembled *in-vitro*-transcripted CymMV-RNA or plasmid DNA in MMG solution (15 mM MgCl_2_, 0.4 M mannitol and 4 mM MES (PH = 5.7)). Transfected protoplasts were incubated at 23 °C for 6–24 h in the darkness. Transfection efficiency was measured according to the expression of the GFP reporter of the transient expression vector pAN580-GFP. The GFP fluorescence was observed and 3–5 images were taken in random distribution under a LSM 710 confocal laser scanning microscope. For protein subcellular localization, the transfected protoplasts were stained with 50 μg/mL 4′-6′-diamidino-2-phenylindole (DAPI) to detect cell nucleus [73]. DAPI signals were excited with Blue Diode Laser under a LSM 710 confocal laser microscope.

### 4.8. RT-PCR and qRT-PCR

The replication of CymMV in transfected *Cymbidium* protoplasts was detected by RT-PCR. Protoplasts transfected with CymMV-RNA were harvested at 12, 24 and 36 h post-transfection (hpt). The first-strand cDNA was obtained from the total RNA of transfected protoplasts and virus-infected *N. benthamiana* leaves. Gene-specific primers were designed according to the CDS of CymMV-coat protein (CymMV-CP) using Primer Premier 5.0 software (Premier, Palo Alto, CA, USA), and *CsUbiquitin* in *Cymbidium* (referred to as *CsUBQ*, Gene bank accession No: AY907703) was used as an internal reference control (Appendix A). RT-PCR was performed in a 20-μL reaction volume comprising 2.0 μL (approximately 50 ng) of first-strand cDNA, 0.8 μL of each forward and reverse primer (10.0 μM), 10.0 μL of 2× Taq Master Mix (Vazyme Biotech Co., Ltd., Nanjing, China) and 6.4 μL of sterile distilled H_2_O. All reactions were performed in 200 μL centrifuge tubes using a Bio-Rad T100™ Thermal Cycler (Bio-Rad, Hercules, CA, USA). The following PCR conditions were used: 95 °C for 5 min, followed by 30 cycles at 95 °C for 15 s, 60 °C for 30 s and 72 °C for 30 s, and then at 68 °C for 5 min. Finally, PCR products were evaluated using agarose gel (2%) electrophoresis. 

The expression levels of *CsNPR1* and *CsPR1* genes were measured by qRT-PCR. Gene-specific primers for qRT-PCR were designed according to CDSs of *CsNPR1* and *CsPR1* genes using the Primer Premier 5.0 software (Premier, Palo Alto, CA, USA) (Appendix A). The gene Cs*UBQ* was used as an internal reference control to normalize the total amount of cDNA in each reaction. PCR was conducted, and only primers that amplified a single product were selected for qRT-PCR. The qRT-PCR was performed in a 20-μL reaction volume comprising 2.0 μL (approximately 20 ng) of 5× diluted first-strand cDNA, 0.8 μL of each primers (10.0 μM), 10.0 μL of 2× SYBR Green I Master Mix (Takara, Dalian, China) and 6.4 μL of sterile distilled H_2_O. All reactions were performed in 96-well reaction plates using a Bio-Rad CFX-96 Real-time PCR System (Bio-Rad, Hercules, CA, USA) with three technical replicates. The following PCR conditions were used: 95 °C for 5 min, followed by 40 cycles at 95 °C for 15 s, 60 °C for 30 s and 72 °C for 30 s, and then at 68 °C for 5 min. The expression of candidate genes was quantified using the relative quantification (2^−ΔΔCT^) method [71]. Each sample was independently collected with three biological replicates.

### 4.9. DAS-ELISA

The presence of CymMV in transfected *Cymbidium* protoplasts was detected by DAS-ELISA using a DAS-ELISA kit (Agdia Inc., Elkhard, IN, USA) following the manufacturer’s instructions. Transfected protoplasts (approximately 1 × 10^6^ protoplasts) were harvested at 24 hpt with three replicates. Leaves from *N. benthamiana* plants infected with CymMV served as positive control, while virus-free *N. benthamiana* leaves served as a negative control. Briefly, protoplasts were centrifuged at 100 g for 2 min, and the supernatant was carefully removed. The protoplast pellets were homogenized using 350 μL general extraction buffer, and then 100 μL of the respective mixture was added into three repeat wells of a 96-well ELISA-plate. The plate was coated with 100 μL coating antibody diluted in coating buffer (1:100, *v/v*) four hours before, and then washed with washing buffer for 4–6 times. After incubation at 4 °C overnight, the wells were washed for 6-8 times using the washing buffer, followed by the addition of the detection antibody and alkaline phosphatase enzyme conjugate (1:100, *v/v*) solution, and incubated at 24 °C for 2 h. The wells were washed 6–8 times using the washing buffer followed by adding 100 μL substrate solution containing 1 mg/mL disodium 4-nitrophenyl phosphate salt. The plate was incubated at room temperature for 30-120 min. The presence of the virus was indicated by a yellow coloration in the respective wells upon visual observation. The absorbance values were measured at 405 nm (OD_405 nm_ value) using a Bio-Rad iMark^TM^ microplate absorbance reader (Bio-Rad Laboratories, Hercules, CA, USA). The presence of the virus was confirmed when the optical density (OD_405 nm_) value was at least five times higher than that of buffer-control wells [74]. 

### 4.10. Statistical analysis

The statistical analysis was performed with SPSS Version 18.0 software (SPSS Inc., Chicago, IL, USA). All experiments were replicated three times. Data are presented as mean–standard error from three independent experiments. Significant differences among treatments were determined at *p* ≤ 0.05 and *p* ≤ 0.01 based on the least significant difference test by Mann–Whitney U-test.

## 5. Conclusions 

Based on a genome-widely identification, three Cs*NPR1-like* (*CsNPR1-1*, *CsNPR1-2* and *CsNPR1-3*) and two *CsPR1-like* (*CsPR1-1* and *CsPR1-2*) genes were isolated from *Cymbidium* orchids. Proteins CsNPR1-1 and CsNPR1-2 were grouped closest to NPR1 homologs in *Zea mays* (sharing 81.98% identity) and *Phalaenopsis* (64.14% identity), while CsNPR1-3 was classified into a distinct group with the *Calotropis procera* NPR1 (70.68% identity) and *Oryza sativa* NPR3 (57.72% identity). Putative CsPR1-1 and CsPR1-2 proteins were highly conserved (99.40% identity) and grouped closest to PR1 proteins in *Phalaenopsis* and other monocot species. Genes *CsNPR1* and *CsPR1* were highly expressed in stem/pseudobulb and/or flower, and they were greatly up-regulated by SA, ROS and CymMV infection, but not by ABA or salinity stress. Compared with *CsNPR1-1* and *CsNPR1-3*, *CsNPR1-2* showed greater fold changes, but *CsPR1-1* and *CsPR1-2* showed almost equal expression levels. In addition, the presence of SA led to the nuclear translocation of CsNPR1-2 protein, and the transient expression of *CsNPR1-2* greatly enhanced expression of *CsPR1* genes. All these results indicate that the *NPR1*-mediated signaling pathway in *Cymbidium* orchids is SA-dependent and can be triggered by ROS induction and CymMV infection. Therefore, the findings of this study can do a great deal to accelerate the healthy growth of orchids, one of the most loved aesthetic beauties the world over. 

## Figures and Tables

**Figure 1 ijms-21-01977-f001:**
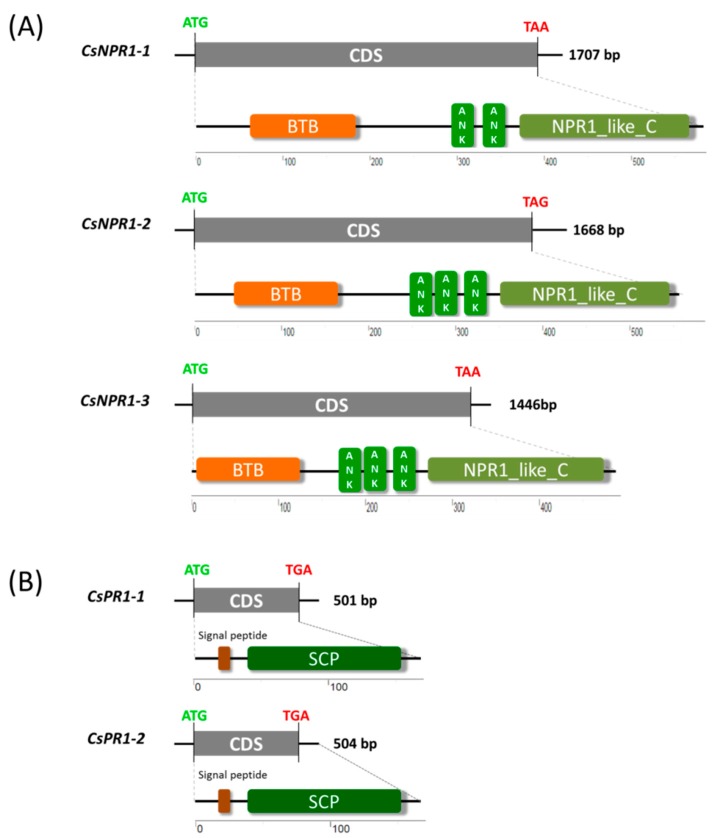
Domain architectures of putative CsNPR1 and CsPR1 proteins in *Cymbidium sinense*. (**A**) Putative CsNPR1 proteins share a conserved broad-complex, tramtrack, bric-a-brac/poxvirus and zinc finger (BTB/POZ) domains, two or three ankyrin repeat (ANK) domains, and a NPR1-like defense protein C domain with other NPR1 homologs; (**B**) CsPR1 proteins share a conserved signal peptide (SP) and a sterol carrier protein (SCP) domain with PR1-like proteins in other plant species.

**Figure 2 ijms-21-01977-f002:**
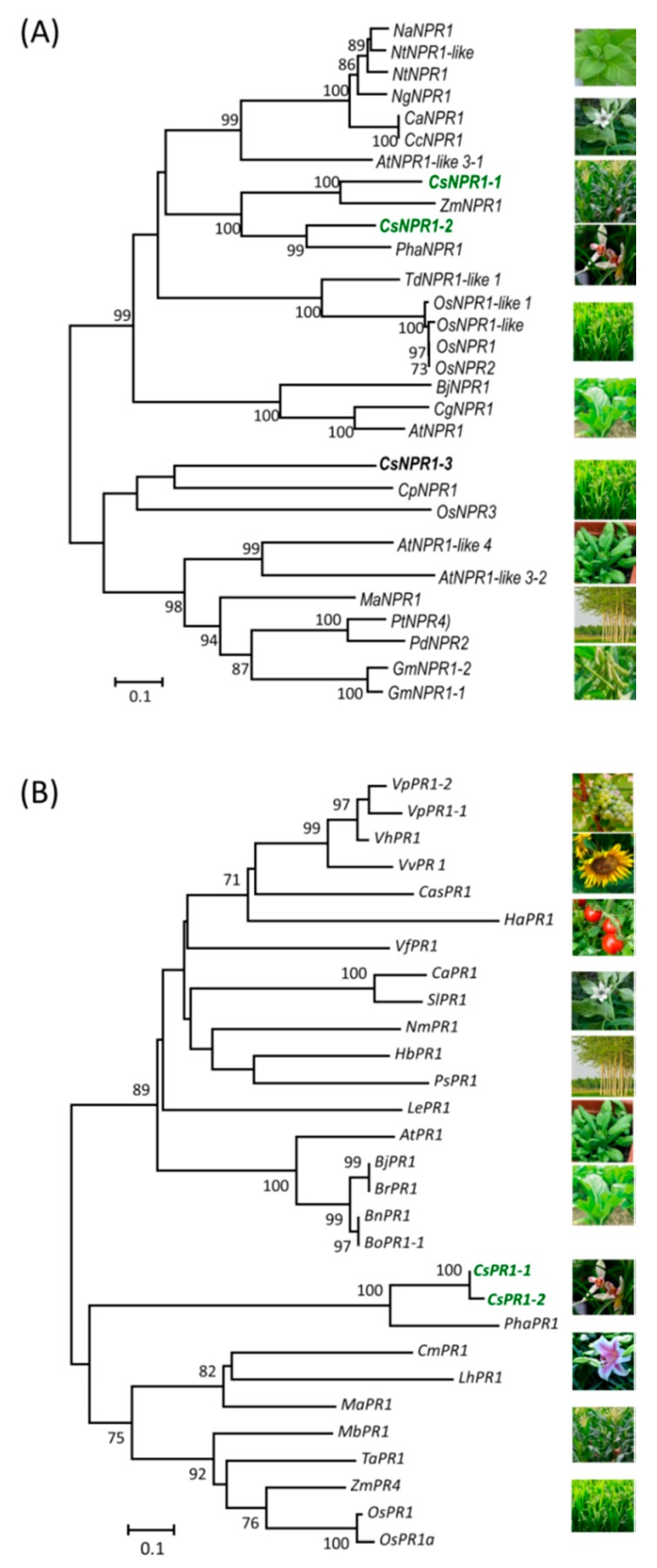
Phylogenetic analyses of *CsNPR1* and *CsPR1* genes and their homologous in various plant species. The deduced amino acid sequences of (**A**) NPR1 and (**B**) PR1 homologs from different plant species were used to construct the phylogenetic trees using the neighbor-joining method with the 1000 bootstrap values indicated. The information for the homologs used in the analysis is listed in Appendix A.

**Figure 3 ijms-21-01977-f003:**
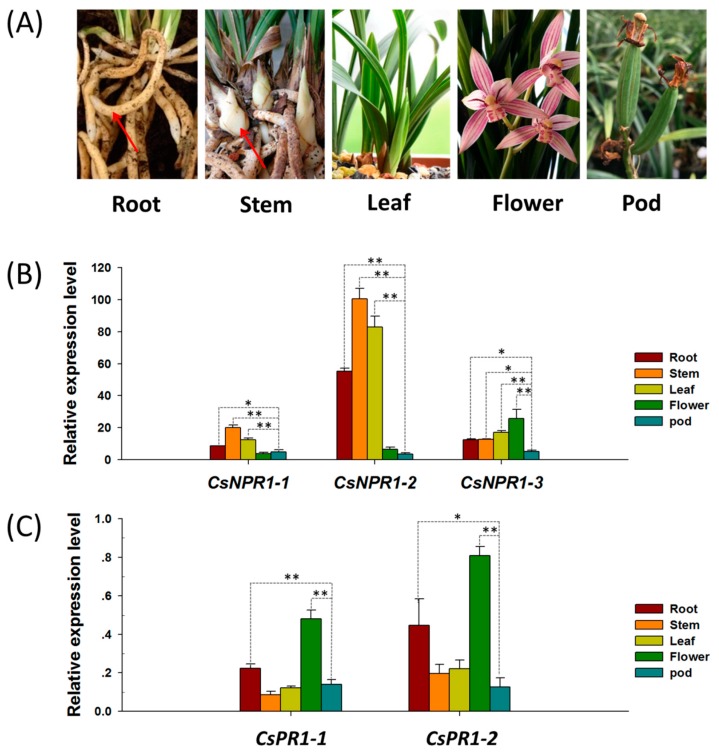
Tissue/organ specific expression analysis of *CsNPR1* and *CsPR1* genes. (**A**) Different tissues/organs (root, stem, leaf, flower and pod) were collected for total RNA isolation; and the expression patterns of (**B**) *CsNPR1* and (**C**) *CsPR1* genes were examined by qRT-PCR. Y-axes indicate the relative expression levels; significant difference was assessed by Mann–Whitney U-test and indicated by asterisks; single asterisk (*) represents *p* ≤ 0.05, double asterisk (**) represents *p* ≤ 0.01; Data are expressed as the mean of three biological replicates, with error bars indicating the SD (standard deviation).

**Figure 4 ijms-21-01977-f004:**
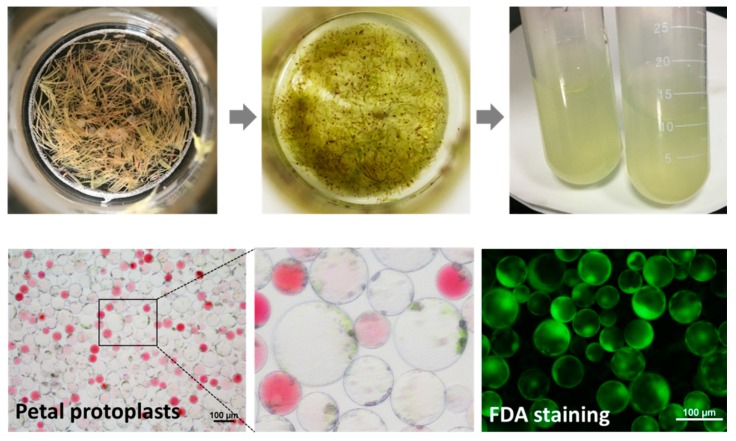
Protoplast isolation from the flower petals of *Cymbidium* orchids.

**Figure 5 ijms-21-01977-f005:**
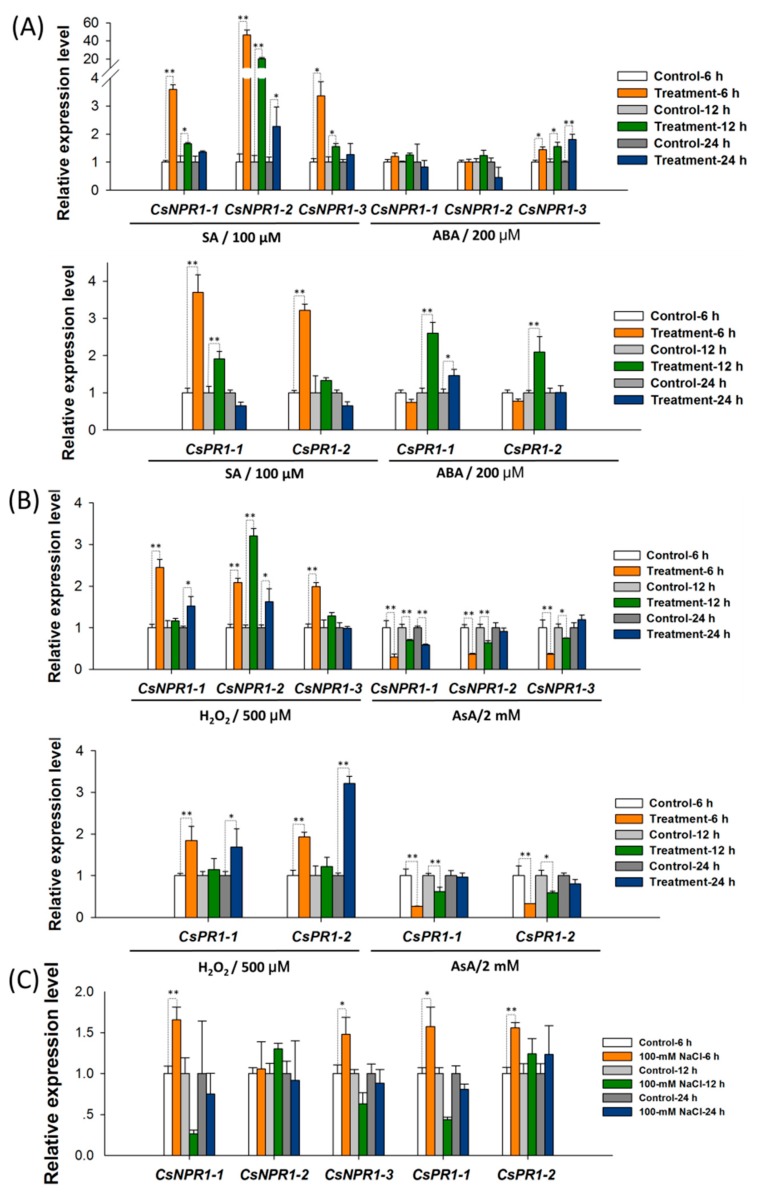
Expression analysis of *CsNPR1* and *CsPR1* genes in response to SA, ABA, H_2_O_2_, AsA and salinity stress induction by qRT-PCR. *Cymbidium* protoplasts were treated with (**A**) SA and ABA, (**B**) H_2_O_2_ and AsA, as well as (**C**) abiotic stresses (NaCl, sodium chloride for salinity stress), and the relative expression levels of *CsNPR1* and *CsPR1* genes were measured by qRT-PCR. Y-axes indicate the relative expression levels; significant difference was assessed by Mann–Whitney U-test and indicated by asterisks; single asterisk (*) represents *p* ≤ 0.05, double asterisk (**) represents *p* ≤ 0.01; Data are expressed as the mean of three biological replicates, with error bars indicating the SD.

**Figure 6 ijms-21-01977-f006:**
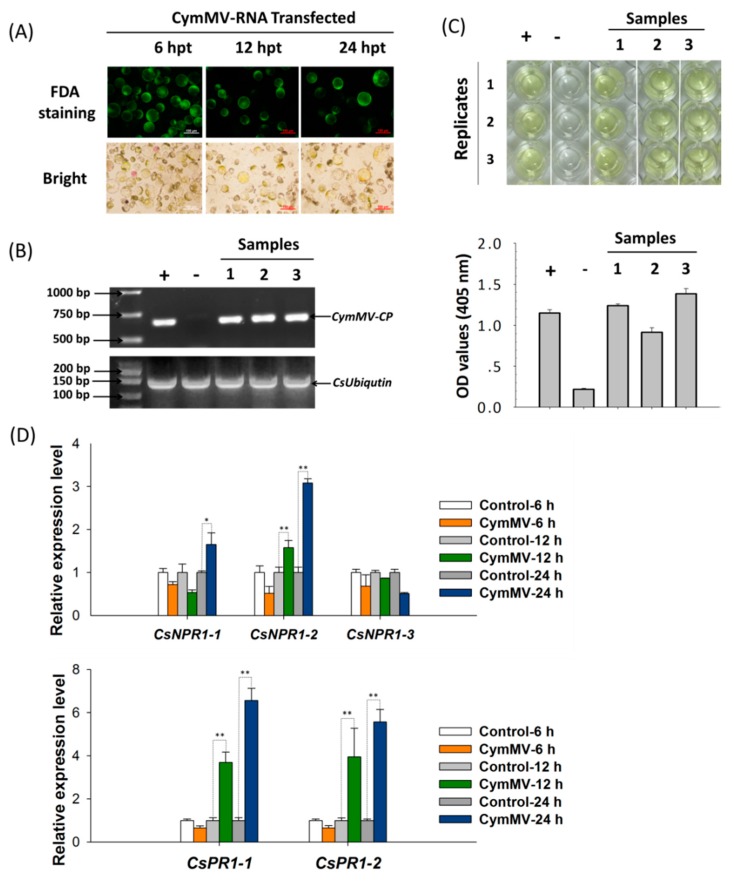
Expression analyses of *CsNPR1* and *CsPR1* genes in response to CymMV infection by qRT-PCR. (**A**) *Cymbidium* protoplasts were transfected with the in vitro transcripts of entire CymMV-RNA, and the viability of transfected protoplasts were examined by FDA staining; (**B**) the replication of CymMV in *Cymbidium* protoplasts were confirmed by RT-PCR with three replicates; (**C**) the replication of CymMV in Cymbidium protoplasts was confirmed by DAS-ELISA with three replicates; (**D**) the relative expression levels of *CsNPR1* and *CsPR1* genes in transfected protoplasts were measured by qRT-PCR; Y-axes indicate the relative expression levels; significant difference was assessed by Mann–Whitney U-test and indicated by asterisks; single asterisk (*) represents *p* ≤ 0.05, double asterisk (**) represents *p* ≤ 0.01; data are expressed as the mean of three biological replicates, with error bars indicating the SD.

**Figure 7 ijms-21-01977-f007:**
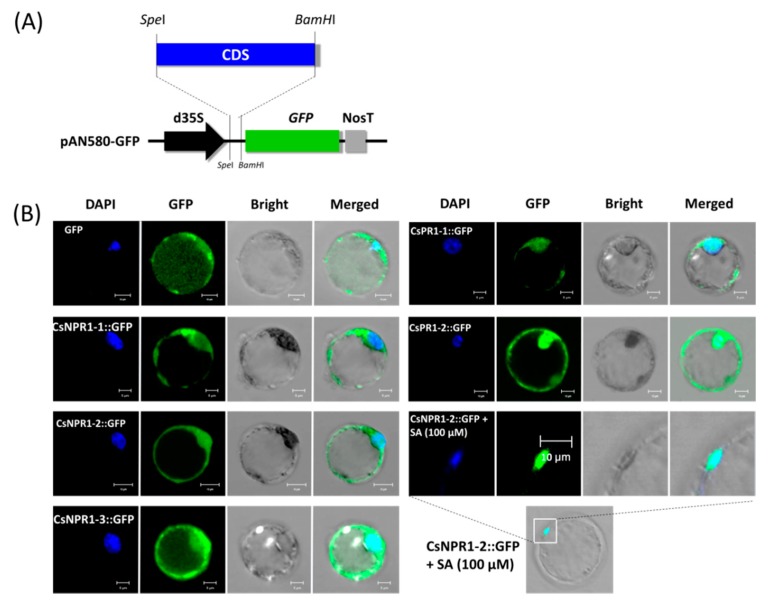
Transient expression of *CsNPR1-2* in *Cymbidium* protoplasts. (**A**) Vectors used for transient expression were obtained by cloning the full-length CDSs into the vector PAN580-GFP; (**B**) Plasmid DNA expressing GFP and fusion protein were transfected into *Cymbidium* protoplasts, and the fluorescence was detected under an LSM 710 confocal laser microscope with a blue excitation block at 12-18 h post-transfection; In the presence of SA (100 μM), fusion protein CsNPR1-2::GFP was co-localized with the DAPI signal, indicating the nuclear localization of CsNPR1-2 protein.

**Figure 8 ijms-21-01977-f008:**
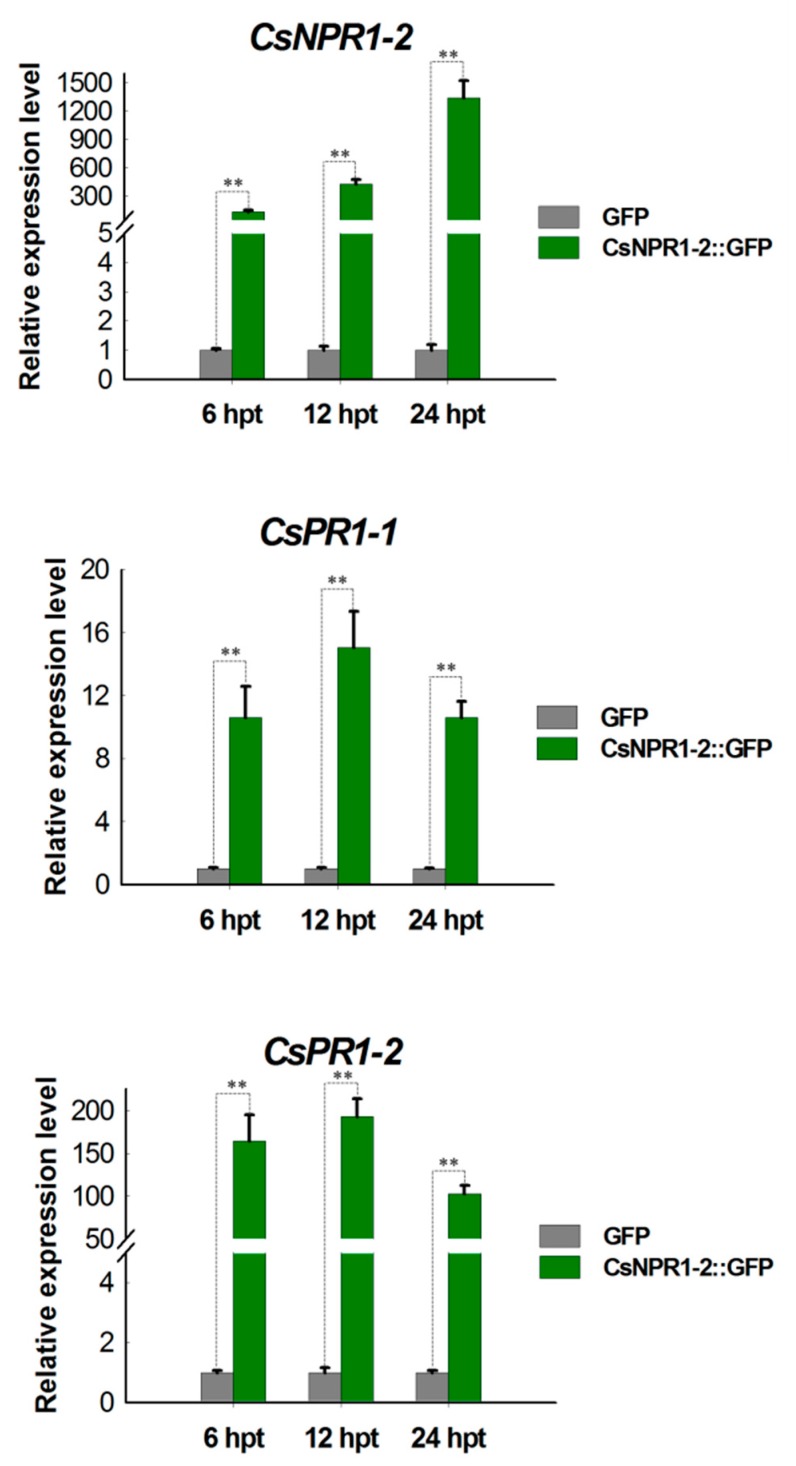
Regulation analysis of *CsNPR1-2* protein on expression of *CsPR1* genes by qRT-PCR. Protoplasts transiently expressing fusion protein CsNPR1-2::GFP were collected at 6, 12 and 24 hpt, and the expression of *CsPR1-1* and *CsPR1-1* genes were examined by qRT-PCR. Y-axes indicate the relative expression levels; significant difference was assessed by Mann–Whitney U-test and indicated by asterisks; single asterisk (*) represents *p* ≤ 0.05, double asterisk (**) represents *p* ≤ 0.01; data are expressed as the mean of three biological replicates, with error bars indicating the SD.

**Figure 9 ijms-21-01977-f009:**
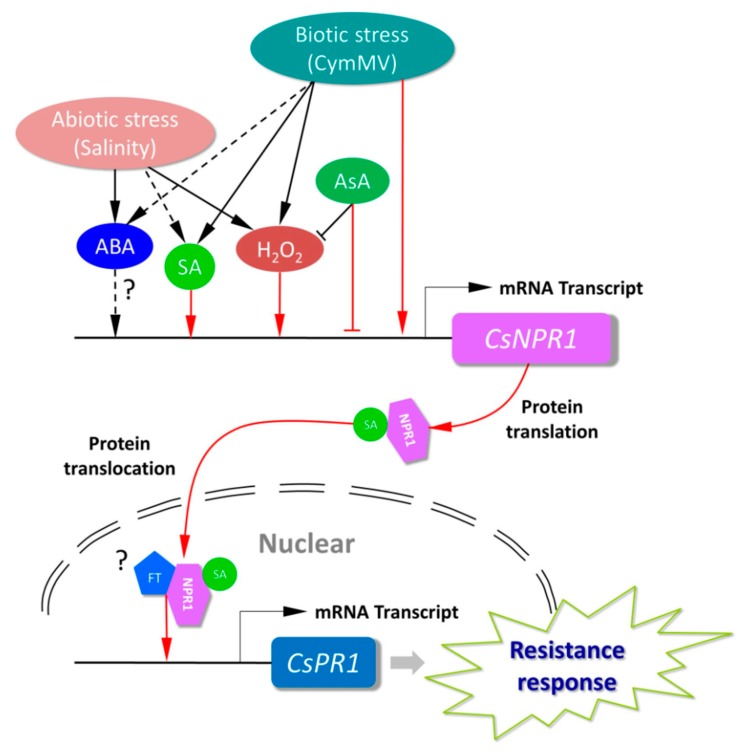
NPR1-mediated signaling pathway in *Cymbidium* orchid. Stress and phytohormones are represented by ellipses; genes are represented by rectangles; and lines before rectangles represent the transcriptional regulatory regions. Lines with an arrow represent the promotion of mRNA transcript, and those with a perpendicular bar represent repression. Red lines indicate the regulation or transportation demonstrated in this study.

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
