# Peer review of "Identification and Characterization of NPR1 and PR1 Homologs in Cymbidium orchids in Response to Multiple Hormones, Salinity and Viral Stresses"

_ijms, 2020, doi:10.3390/ijms21061977_

Round 1

Reviewer 1 Report

Title: The CsNPR1-2 -mediated signaling pathway in Cymbidium orchids is Salicylic acid-dependent and triggered by CymMV infection

In this manuscript, the genome widely identified the three homologous of NPR1 and two PR1 in Cymbidium orchids and analyzed the expression of these genes in different organs. Additionally, the authors also analyzed the expressions of these genes in responses to different hormones, H2O2, ascorbic acids as well as the viral and salinity stresses. I appreciate the author's approach that they try to understand the molecular-level changes in the relatively less important crops like Cymbidium orchids in responses to different stimuli. However, the hypothesis and the significance of the obtained results have not been expressed clearly. The authors need to substantially improve the manuscript to be acceptable for the publication.

Issues to be addressed:

Title:

The title does not reflect the hypothesis and/or objectives of the current work.

Abstract:

The background of the abstract not clear, authors are suggested to present the background clearly.

Line 15: PR1 is widely known as Pathogenesis-related 1, change the “pathogen-related” to “pathogenesis-related”.

Line21: Also change the “hydrogen dioxide’ to “hydrogen peroxide”.

Introduction:

The logic of this manuscript is difficult to follow. For instance, in the second paragraph of the introduction authors describe the plant immunity system, however, in whole manuscript I didn’t find any relationship or data regarding PTI and/or ETI. In the introduction, authors should follow the logic flow which helps the readers to get a clear idea about the hypothesis and objectives of this study.

Materials and methods:

Methods used for the statistical analysis should be presents in Materials and methods.

Lines 149-158: Did the authors conducted the pretest to determine the optimum concentration of different treatments? Or describe the justification for this.

Results:

Overall the results should present more precisely; For example, the gene expression significantly up- or down-regulated compared to what?

In figure 3C: What are the meaning of the significance and they depressed in all the organs?

In figures 4B, 5 and 6D: Authors determined the relative expression by using the 0h as control or control of each time point set as an internal control to analyze the relative expressions by using (2-ΔΔCT), please indicates clearly.

In figure 4B: Why only 6h after SA treatment showed the highest expression?

Also I would suggest to present the both hormones data together (SA, ABA) in figure 4.

In figure 5A: Authors need to check carefully the meaning of the significance which is very much doubtful in AsA treatment?

Discussion:

Authors only present the effects of different hormones and other stresses in the protoplast of Orchids. However, if Authors can treat or imposed the stress (especially salinity; that is much convenient) in the whole plants and analyzed the responses of theses homologs of NPR1 and PR1 genes in relation to stress symptom development and/or stress acclimation process that will improved our knowledge and beneficial to adapts this phenomenon for management of the diverse biotic and abiotic stresses in Orchids.

Author Response

#Reviewer 1

Comment 1: In this manuscript, the genome widely identified the three homologous of NPR1 and two PR1 in Cymbidium orchids and analyzed the expression of these genes in different organs. Additionally, the authors also analyzed the expressions of these genes in responses to different hormones, H2O2, ascorbic acids as well as the viral and salinity stresses. I appreciate the author's approach that they try to understand the molecular-level changes in the relatively less important crops like Cymbidium orchids in responses to different stimuli. However, the hypothesis and the significance of the obtained results have not been expressed clearly. The authors need to substantially improve the manuscript to be acceptable for the publication.

Answer: We have addressed all the issues and revised the manuscript carefully. The changes made in the manuscript are highlighted and the itemized list of responses to the comments is as following.

Issues to be addressed:

Comment 2: The title does not reflect the hypothesis and/or objectives of the current work.

Answer: According to the hypothesis and/or objectives of our work, the title of the article has been modified from “The CsNPR1-2 -mediated signaling pathway in Cymbidium orchids is Salicylic acid-dependent and triggered by CymMV infection” to “Identification and characterization of NPR1 and PR1 homologs in Cymbidium orchids in response to multiple hormones, salinity and viral stresses”.

Comment 3: The background of the abstract is not clear, and authors are suggested to present the background clearly.

Answer: According to reviewer’s comment, the background of the abstract has been rewritten as following “The plant nonexpressor of pathogenesis-related 1 (NPR1) and pathogenesis-associated 1 (PR1) genes play fundamental roles in plant immunity response, as well as abiotic-stress tolerance. Nevertheless, comprehensive identification and characterization of NPR1 and PR1 homologs had not been conducted so far in Cymbidium orchids, a valuable industrial crop cultivated as ornamental and medicinal plants worldwide.” (Abstract section, Line 15-19, in the revised manuscript)

Comment 4: Line 15: PR1 is widely known as Pathogenesis-related 1, change the “pathogen-related” to “pathogenesis-related”.

Answer: According to reviewer’s suggestion, we have changed the “pathogen-related” into “pathogenesis-related”. (Abstract, Line 15 in the revised manuscript)

Comment 5: Line21: Also change the “hydrogen dioxide’ to “hydrogen peroxide”.

Answer: According to reviewer’s comment, “hydrogen dioxide” has been changed into “hydrogen peroxide”. (Abstract, Line 29, in the revised manuscript)

Comment 6:

The logic of this manuscript is difficult to follow. For instance, in the second paragraph of the introduction, authors describe the plant immunity system, however, in whole manuscript I didn’t find any relationship or data regarding PTI and/or ETI. In the introduction, authors should follow the logic flow which helps the readers to get a clear idea about the hypothesis and objectives of this study.

Answer: According to reviewer’s comment, the section of the introduction was carefully revised. We removed the description of plant immunity system and improved the logic of the introduction and the whole manuscript.

Comment 7: Methods used for the statistical analysis should be presents in Materials and methods.

Answer: Methods used for the statistical analysis have been added in “Materials and methods” as following: “All experiments were replicated three times. The statistical analysis was performed with SPSS Version 18.0 software (SPSS Inc., Chicago, IL, USA). Data are presented as mean -standard error from three independent experiments. Significant differences among treatments were determined at p ≤ 0.05 and p ≤ 0.01 based on the least significant difference test by Mann-Whitney U-test”. (Methods, 4.10; Line 693-697, in the revised manuscript)

Comment 8: Lines 149-158: Did the authors conducted the pretest to determine the optimum concentration of different treatments? Or describe the justification for this.

Answer: Because of there is little reference for different hormones, H2O2/AsA and salinity stress treatments in orchids, we have conducted the pretest to determine the optimum concentrations of different treatments. The gradient concentrations of different treatments in this study was set (data not shown), according to the optimal concentration of different treatments in plants, such as Arabidopsis (Kovacs et al., 2015; DOI: https://nph.onlinelibrary.wiley.com/doi/full/10.1111/nph.13502), rice (Hwang and Hwang, 2010; https://link.springer.xilesou.top/article/10.1007/s11816-009-0116-5), Phalaenopsis (Chen et al., 2013; https://as-botanicalstudies.springeropen.com/articles/10.1186/1999-3110-54-31) and soybean (Rui et al., 2017; https://link.springer.xilesou.top/article/10.1007/s00122-017-2966-5) (Methods, 4.5; Line 605-607, in the revised manuscript). For example, protoplasts were treated with 1, 10, 100 and 200 -μM SA supplemented in W5 solution, and then the relative expression levels of NPR1 genes were examined at 6, 12 and 24 hpt by qRT-PCR. Compared with other concentrations, 100 -μM SA causes maximum upregulation of NPR1 genes. Hence, 100 -μM were selected as the optimum concentration of SA to orchid protoplasts. In the same way, the optimum concentration of H2O2, AsA, ABA and NaCl were determined as 500-μM, 2-mM, 200-μM and 100-mM, respectively.

Comment 9: Overall the results should present more precisely; For example, the gene expression significantly up- or down-regulated compared to what?

Answer: We have revised the manuscript carefully and tried our best to describe the results precisely. The genes’ expressions significantly up or down -regulated were compared to the control of each time point. Moreover, we modified figures, and added comparison symbols for significant difference.

Comment 10: In figure 3C: What are the meaning of the significance and they depressed in all the organs?

Answer: For tissues/organs-specific expression analysis of CsNPR1 and CsPR1 genes, the related expression levels of these genes in root, stem, leaf and flower were all compared with that of pod. The significance only means the differences between the expression levels of these genes in root, stem, leaf and flower, when compared with that of pod. The related expression levels were calculated with 2-ΔΔCT method, without normalization of control (usually the normalized expression levels of control were close to 1.00). In this way, we can compare the relative expression levels between genes. The related expression levels of CsPR1 genes were quite lower than that of CsNPR1 genes (figure 3B, 3C).

Comment 11: In figures 4B, 5 and 6D: Authors determined the relative expression by using the 0h as control or control of each time point set as an internal control to analyze the relative expressions by using (2-ΔΔCT), please indicates clearly.

Answer: In figures 4B, 5 and 6D, we determined the relative expression by using the control of each time point set as control (Line 343-404, in the revised manuscript). The related expression levels were calculated with 2-ΔΔCT method with the normalized expression levels of control (close to 1.00…..). We have revised relevant results all through the manuscript.

Comment 12: In figure 4B: Why only 6h after SA treatment showed the highest expression?

Answer: Protoplasts are plant cells whose cell wall has been removed. They can directly, rapidly and sensitively perceive extracellular stimulates, and transduce the related signals to intracellular via receptors on cytomembrane. The functions predicted for genes up-regulated by SA allowed us to separate these genes into 2 main groups: genes involved in defense against stress; and genes involved in signal transduction. NPR1-dependent and independent genes were reported to be early (in 300 min) induced by salicylic acid treatment in Arabidopsis. (Blanco et al., 2005; DOI: https://link.springer.xilesou.top/article/10.1007/s11103-005-2227-x). Therefore, it is reasonable that the gene expressions increased rapidly and showed the highest expression at 6 hpt, and then decreased gradually as time goes on.

Comment 13:

Also I would suggest to present the both hormones data together (SA, ABA) in figure 4.

Answer: The both hormones (SA, ABA) data were put together in “figure 5A” according to reviewer’s suggestion.

Comment 13: In figure 5A: Authors need to check carefully the meaning of the significance which is very much doubtful in AsA treatment?

Answer: We are really sorry for the insufficient description of this part, resulting to confusion of understanding the significance differences between AsA treatment and control at each time points. In the modified “Figure 5”, we have added mandatory signs for the significance differences to address this issue. As shown in modified “Figure 5” and described in revised “Results 2.4” (Line 343-362, in the revised manuscript), “Expressions of CsNPR1 and CsPR1 genes were significantly up-regulated in the presence of H2O2, but were remarkably inhibited  with added AsA when compared with controls at each time points”.

Comment 14: Discussion: Authors only present the effects of different hormones and other stresses in the protoplast of Orchids. However, if Authors can treat or imposed the stress (especially salinity; that is much convenient) in the whole plants and analyzed the responses of theses homologs of NPR1 and PR1 genes in relation to stress symptom development and/or stress acclimation process that will improved our knowledge and beneficial to adapts this phenomenon for management of the diverse biotic and abiotic stresses in Orchids.

Answer: Many thanks for reviewer’s suggestion on our research work in stress symptom development and/or stress acclimation process. Actually, we have done hormone and salinity treatment with plant in preliminary experiment, using 100-μM SA, 200-μM ABA, 500-μM H2O2 2-mM AsA and 100-mM NaCl, and got 2 to10-fold change of the marker genes such as NPR1 and PR1 (unpublished data). However, as described in “introduction” section, “the long vegetative growth period of orchids” makes it very difficult to get enough plant materials of the same growth period in a specific cymbidium orchid plant. So we recently developed an efficient protoplast system for gene functional characterization. Compared with plant treatments, protoplast treatments can be simultaneously carried out, which is more repeatable and precise with the same trend as that of plant leaf treatment. In our future reports, we will systemically discuss the benefit of our protoplast system in hormone and stress signal transduction in orchid.

Reviewer 2 Report

The article of Yonglu et al. describes the identification and characterization of NPR1 and PR1 homologs in Cymbidium orchids. The identification and phylogenetic analysis of these genes reveal distinct relationship. The expression profile of the genes were carried out in protoplast system in the presence of SA, H2O2, ABA, NaCl and Cymbidium mosaic virus. In this section the results needs improvement, especially in line 287-293. According to the line 287 “the expression levels of CsNPR1-1 and CsNPR1-2 were higher in stem and flower”, but according to the Fig 3B the expression level is the higher in stem and leaf in these cases, and in the CsNPR1-2 the expression level is also high in root- but not in the flower. According to the figure 3B the expression pattern of the CsNPR1-1 and CsNPR1-3 is not similar, but the expression level comparable.

In the case of Fig 7 I suggest to include the photo of all the fusion protein constructs in the main text to emphasize the different location. Both Fig 7B and Fig 7C represents GFP and CsNPR1-2:::GFP, so I suggest to remove panel B.

Generally the article is interesting but the result section needs reconsideration.

Author Response

#Reviewer 2

The article of Yonglu et al. describes the identification and characterization of NPR1 and PR1 homologs in Cymbidium orchids. The identification and phylogenetic analysis of these genes reveal distinct relationship. The expression profile of the genes were carried out in protoplast system in the presence of SA, H2O2, ABA, NaCl and Cymbidium mosaic virus. Generally the article is interesting but the result section needs reconsideration.

Comment 1: In this section the results needs improvement, especially in line 287-293. According to the line 287 “the expression levels of CsNPR1-1 and CsNPR1-2 were higher in stem and flower”, but according to the Fig 3B the expression level is the higher in stem and leaf in these cases, and in the CsNPR1-2 the expression level is also high in root- but not in the flower.

Answer: Many thanks for your detailed comments and corrections, and sorry for our carelessness. In the updated version, the section of the results was revised carefully according to the comment. Thereinto, results of “Tissues/Organs-Specific Expression Analysis of CsNPR1 and CsPR1 Genes” were revised according to the modified figure 3B. We miswrote “leaf” as “flower”, and revised the sentence “the expression levels of CsNPR1-1 and CsNPR1-2 were higher in stem and flower” to “CsNPR1-1 and CsNPR1-2 showed similar expression patterns regardless of their overall gene expression levels, with the highest expression levels in stem/pseudobulb followed by flower” (Line 311-313, in the revised manuscript).

Comment 2: According to the figure 3B the expression pattern of the CsNPR1-1 and CsNPR1-3 is not similar, but the expression level comparable.

Answer: We rechecked the relevant results, and found the expression pattern of the CsNPR1-1 and CsNPR1-3 is indeed not similar. We meant to express that CsNPR1-1 and CsNPR1-2 have similar expression pattern but CsNPR1-2 is dominant in all plant organs compared with that of CsNPR1-1. However, we wrote CsNPR1-2 as 1-3 due to typing error. The expression pattern of the CsNPR1-1 and CsNPR1-3 is indeed not similar. We have corrected the description in the revised manuscript accordingly “The expression patterns of CsNPR1-1 and CsNPR1-2 were similar, but the relative expression levels of CsNPR1-1 and CsNPR1-3 were much lower than that of CsNPR1-2 overall, which indicated that the active CsNPR1-2 might have a main role.” (Line 314-317, in the revised manuscript).

Comment 3: In the case of Fig 7 I suggest to include the photo of all the fusion protein constructs in the main text to emphasize the different location. Both Fig 7B and Fig 7C represents GFP and CsNPR1-2:::GFP, so I suggest to remove panel B.

Answer: According to reviewer’s comment, all the fusion protein constructs were put together in the “Figure 7” in the main text; the repeated figures represent GFP and CsNPR1-2:::GFP “Figure 7 B” were removed.

Round 2

Reviewer 1 Report

The authors addressed all the points raised by my side. The current form of the manuscript is well written and easy to understand and is suitable for publication in IJMS.

Reviewer 2 Report

The proposed modifications were carried out on the MS, I tink that it's present form is adequate for publication.

This manuscript is a resubmission of an earlier submission. The following is a list of the peer review reports and author responses from that submission.